# Rosette-Forming Glioneuronal Tumor of the Fourth Ventricle: A Case of Relapse Treated with Proton Beam Therapy

**DOI:** 10.3390/diagnostics11050903

**Published:** 2021-05-19

**Authors:** Antonella Cacchione, Angela Mastronuzzi, Andrea Carai, Giovanna Stefania Colafati, Francesca Diomedi-Camassei, Antonio Marrazzo, Alessia Carboni, Evelina Miele, Lucia Pedace, Marco Tartaglia, Maurizio Amichetti, Francesco Fellin, Mariachiara Lodi, Sabina Vennarini

**Affiliations:** 1Department of Paediatric Haematology/Oncology, Cell and Gene Therapy, Bambino Gesù Children’s Hospital, Istituto di Ricovero e Cura a Carattere Scientifico, 00165 Rome, Italy; angela.mastronuzzi@opbg.net (A.M.); evelina.miele@opbg.net (E.M.); lucia.pedace@opbg.net (L.P.); mariachiara.lodi@opbg.net (M.L.); 2Neurosurgery Unit, Department of Neuroscience and Neurorehabilitation, Bambino Gesù Children’s Hospital, Istituto di Ricovero e Cura a Carattere Scientifico, 00165 Rome, Italy; andrea.carai@opbg.net; 3Oncological Neuroradiology Unit, Department of Imaging, Bambino Gesù Children’s Hospital, Istituto di Ricovero e Cura a Carattere Scientifico, 00165 Rome, Italy; gstefania.colafati@opbg.net (G.S.C.); antonio.marrazzo@opbg.net (A.M.); alessia.carboni@opbg.net (A.C.); 4Department of Laboratories, Pathology Unit, Bambino Gesù Children’s Hospital, Istituto di Ricovero e Cura a Carattere Scientifico, 00165 Rome, Italy; francesca.diomedi@opbg.net; 5Genetics and Rare Diseases Research Division, Bambino Gesù Children’s Hospital, Istituto di Ricovero e Cura a Carattere Scientifico, 00165 Rome, Italy; marco.tartaglia@opbg.net; 6Proton Therapy Center, Hospital of Trento, Azienda Provinciale per I Servizi Sanitari (APSS), 38123 Trento, Italy; maurizio.amichetti@apss.tn.it (M.A.); francesco.fellin@apss.tn.it (F.F.); sabina.vennarini@apss.tn.it (S.V.)

**Keywords:** pediatric brain tumor, Rosette-forming glioneuronal tumors, relapse, proton beam therapy

## Abstract

Rosette-forming glioneuronal tumors (RGNTs) are rare, grade I, central nervous system (CNS) tumors typically localized to the fourth ventricle. We describe a 9-year-old girl with dizziness and occipital headache. A magnetic resonance imaging (MRI) revealed a large hypodense posterior fossa mass lesion in relation to the vermis, with cystic component. Surgical resection of the tumor was performed. A RGNT diagnosis was made at the histopathological examination. During follow-up, the patient experienced a first relapse, which was again surgically removed. Eight months after, MRI documented a second recurrence at the local level. She was a candidate for the proton beam therapy (PBT) program. Three years after the end of PBT, the patient had no evidence of disease recurrence. This report underlines that, although RGNTs are commonly associated with an indolent course, they may have the potential for aggressive behavior, suggesting the need for treatment in addition to surgery. Controversy exists in the literature regarding effective management of RGNTs. Chemotherapy and radiation are used as adjuvant therapy, but their efficacy management has not been adequately described in the literature. This is the first case report published in which PBT was proposed for adjuvant therapy in place of chemotherapy in RGNT relapse.

## 1. Introduction

Rosette-forming glioneuronal tumor (RGNT) is a rare neoplasm. RGNT—classified as a World Health Organization (WHO) grade I glioneuronal neoplasm—has a distinctive histological appearance with mixed glial and neurocytic components [1]. First described by Komori et al., it is a slow-growing tumor, typically localized to the fourth ventricle and composed of neurocytes that form neurocytic rosettes and glial components similar to those of the pilocytic astrocytoma [2]. Due to the rarity of the RGNT, little is known about its epidemiology and natural history. Gross total resection (GTR) is the treatment of choice, and there is no solid evidence supporting other adjuvant treatment options when GTR is not feasible. Few case reports are described in literature about patients with relapse; therefore, there is a non-unanimous agreement on therapeutic strategies for disease recurrence. We report a case of a girl treated at recurrence with proton beam therapy (PBT). PBT offers dosimetric advantages over photon radiation therapy due to steep dose fall-off at depth. Decreased integral dose to normal brain and superiority in meeting surrounding tissue constraints has established the role of PBT in management of various central nervous system (CNS) tumors. Studies have demonstrated the dosimetric advantages of PBT, allowing for reduction in radiation-induced toxicities, such as neurocognitive decline and secondary malignancy [3,4,5].

## 2. Case Presentation

A nine-year-old girl was referred to the Pediatric Emergency Department for worsening headache episodes that began about a year previously associated with vomiting. Clinical evaluation showed she had no neurological abnormalities. Ophthalmological evaluation was then performed with the finding of signs of intracranial hypertension. Urgent brain computed tomography (TC) was performed, showing the presence of obstructive triventricular hydrocephalus caused by a voluminous median expansive cerebellar lesion (Figure 1A). Brain and spinal magnetic resonance imaging (MRI) confirmed the presence of an intra-axial vermian lesion that came to lick the cistern of the quadrigeminal lamina (Figure 1B–F). 

The patient was then urgently subjected to endoscopic third ventriculocisternostomy neurosurgery with clear improvement of the intracranial hypertension and resolution of headaches and emesis. A few days later, the girl underwent suboccipital craniotomy surgery with macroscopically complete removal of the lesion. The post-operative course took place in the absence of significant complications. A brain MRI performed twenty-four hours after surgery documented the complete removal of the mass (Figure 1G). 

Histological examination revealed a biphasic hypocellular neoplasm, composed of synaptophysin-positive monomorphic small/medium-sized cells with a rounded nucleus and clear cytoplasm, that were focally arranged around small vessels (perivascular pseudorosettes) or around a fibrillar core (neurocytic rosettes). The glial cells consisted of spindle- or stellate-shaped astrocytic cells with elongated to oval nuclei forming a compact fibrillar meshwork with occasional Rosenthal fibers and oligodendroglial-like cells with round nuclei and clearly staining cytoplasm were present. Numerous microcalcifications were present. The proliferation index (anti-Ki67) was about 1%. In light of the identified findings, a diagnosis of low-grade mixed glioneuronal neoplasm with histological characteristics suggestive of the glioneuronal tumor forming rosettes of the IV ventricle (grade I according to WHO) was performed (Figure 2A–C).

Two years after diagnosis, a nodular lesion in the surgical field was discovered on a follow-up MRI (Figure 1H). The patient therefore underwent a second neurosurgery with complete removal of the mass from a macroscopic point of view (Figure 1I). Histological analysis confirmed the recurrence of the same neoplasm (Figure 2D–F).

Eight months after the second surgery, an MRI documented a second recurrence at the local level (Figure 1L). The girl was therefore a candidate for proton therapy program (Figure 1M). The proton treatment was carried-out for one month (Figure 3A–C). Dose distributions of the 3-fields proton plan, optimized with single-field-optimization (SFO) technique to deliver 54 Gy relative biological effectiveness (RBE) in 30 fractions (1.8 Gy RBE per fraction).

Three years after the end of proton therapy, the patient has no evidence of further disease recurrence during the MRI performed at follow-up (Figure 1N). 

## 3. Discussion

RGNT was initially described as a cerebellum dysembryoplastic neuroepithelial tumor (DNT) [6]. Komori et al. first described, in a series of 11 cases, RGNT as a distinct variant of mixed glioneuronal tumor arising in relation to the fourth ventricle [2]. In 2016 it was included as a low grade (Grade I) tumor in the last edition of the WHO classification of CNS tumors [7]. Although rare, RGNTs involving an extra-ventricular site, such as the pineal gland region, optic chiasm, hypothalamus, and even the cervical spinal cord, have been reported [8,9]. It occurs predominantly in young adults with the mean age of about 31.5 years (range: 12–59 years) [10]. There is a female preponderance (F:M ratio around 2:1). No familial association has been described. However, a few cases have been found in association with neurofibromatosis type 1 (NF1), as well as two with Noonan syndrome (NS) [11,12,13]. No genetic link was identified. Most patients have experiences of chronic headaches, ataxia, and emesis, such as other posterior fossa lesions. Other specific symptoms depend on the localization of the tumor.

From a radiological point of view, RGNT presents as iso- to hypo-intense on T1-weighted and hyperintense on T2-weighted MRI images. Calcification may also be present. The tumor is relatively well circumscribed, with both solid and cystic components. The cysts may be single or multiple. Variable degrees of contrast enhancement may also be seen [14]. Secondary hydrocephalus may also be found. Satellite lesions have been detected in a small number of patients. 

Pathological examination showed these tumors have been grossly described to be soft, gelatinous, and generally well demarcated, and sometimes minor to moderate infiltration has been observed. Microscopic examination shows two distinct components. The neurocytic component consists of a uniform population of neurocytes forming neurocytic rosettes and/or perivascular pseudorosettes. Neurocytes are seen to radiate around central neuropil cores and are dispersed in a loose, mucoid, often microcystic manner. Second component is the glial component with spindle and piloid cells resembling pilocytic astrocytoma. Rosenthal fibers and granular bodies are rare. Thrombosis of vessels and evidence of endothelial proliferation have been described in some cases [15]. Significant cytologic atypia and mitotic activity are lacking. On immunostaining, synaptophysin labels, the neuropil matrix of the neurocytic rosettes, and the astrocytic component, are labeled by GFAP and S-100. Although co-expression of neuronal and glial markers in the neurocytic component of RGNT has recently been described [16]. MIB-1 proliferation index is typically low (ranging from 0.35% to 3.07%) [2]. Differential diagnosis of RGNT includes pilocytic astrocytoma, DNT, oligodendroglioma, and central neurocytoma. Pilocytic astrocytoma can be differentiated from RGNT by the absence of rosettes or perivascular pseudorosettes.

The histogenesis of RGNT is still unclear. RGNTs have been postulated to arise from pluripotent cells of the subependymal plate (periventricular germinal matrix) [2,16,17]. However, the alternative possibility of histologically similar tumors with underlying genetic differences cannot be ruled out in light of the increased incidence of RGNT originating beyond the fourth ventricle. Recent papers have described the molecular genetic features of RGNT. Although it may resemble pilocytic astrocytoma, no BRAF alteration, fusion, or mutation has been demonstrated [18]. 

Many genetic mutations have also been found in association with RGNT. The most common genetic marker associated with RGNT is the PIK3CA mutation [19,20,21]. FGFR1 mutations have also been implicated in tumor pathogenesis [21,22], as well as IDH1 [23]. KIAA1549/BRAF gene fusion [24], PPP1R1A, and RNF21 [13] are described as other molecular features. A recent retrospective cohort study was able to obtain tissue samples from 18 different patients with RGNT for comprehensive genomic testing [21]. Sequencing of those samples revealed that all 18 patients contained a FGFR1 mutation, with 13 carrying a comorbid PIK3CA mutation. The study’s findings confirmed that constitutive activation of the FGFR signaling likely plays a role in a large portion of these tumors.

Surgery is the mainstay of treatment. Although RGNTs usually have the possibility of a surgical cure and favorable prognosis, gross total resection (GTR) is typically actuated only if there is very low risk of neurological injury. Subtotal resection (STR), partial resection (PR), and simple biopsy have all been employed [25,26,27]. The most recent review published in literature regarding RGNTs collect 116 adult and pediatric cases [28], among whom, 62 underwent GTR (53%), 32 underwent STR (28%), 7 underwent a partial resection (6%), and 4 underwent only a biopsy (3%). In this series, adjunctive treatments included radiotherapy (9 cases, 8%) and chemotherapy (4 cases, 3%). 

Approximatively 10% is the estimated probability of recurrence when GTR or STR are performed. Wilson P.W. et al. [28] in fact showed 12 cases of recurrence in their review of 116 cases; in the group of 62 patients treated with GTR, after a mean time of 6.1 years, 6 had a recurrence; 3 were the rate in the group of 32 STR; 1 in 7 PR, and 2 in the 4 cases only biopsied. There is not a large enough sample size of recurrences to achieve significant conclusions regarding the most effective treatment plan for them.

In 2017, the largest pediatric series has been collected in the review published by Morris C. et al. [29] with 33 collected cases from literature. Five recurrences have been described in pediatric age: three patients retreated with GTR, one patient with chemotherapy based on temozolomide and cis retinoic acid (intraventricular dissemination) and one 8-year-old treated at recurrence with STR and radiotherapy.

Radiotherapy for pediatric brain tumors is a leading contributor to adverse neurocognitive outcomes, endocrine deficits and contributes to hearing loss [30,31]. Reduction in long-term toxicity will become an essential component of improving quality of life (QoL) in this population. Strategies to reduce late-effects in pediatric cancer survivors have focused on reductions of intensity of radiotherapy treatment. The advent of proton beam therapy (PBT) has emerged as a major paradigm shift in the delivery of radiation therapy for both pediatric and adult patients. The physical properties of protons allow for reduction in radiation exit and integral dose, compared with typical photon plans. Sparing of non-target tissues is a critical aspect of treating pediatric brain tumors in an effort to minimize sequelae such as cognitive dysfunction, endocrinopathies, necrosis, and the risk of secondary malignancies. The St. Jude Lifetime Cohort Study (SJLIFE) showed that there is a correlation between higher radiation therapy doses and worse chronic health conditions [32]. Several studies have shown significant impairments in psychosocial functioning, attention, processing speed and measures of intelligence for survivors of acute lymphoblastic leukemia and medulloblastoma who received radiation therapy [33,34]. As PBT typically reduces dose to normal tissue by a factor of 2–3, and better spares normal brain, the hypothalamic–pituitary axis and cochlea, there is growing interest in leveraging its dosimetric benefits in the pediatric brain tumor population. 

Our patient allows us to reflect on the characteristics of this type of neoplasm, although RGNTs are commonly associated with an indolent course, they may have the potential for aggressive behavior and recurrence, suggesting the need for treatment in addition to surgery. The report suggests about the possibility of evaluating an adjuvant therapy and on the timing of a therapy. 

Our patient’s follow up (three-year) highlights radiological complete remission without any sufferance sign in the surrounding structures and no clinical nor neurological abnormalities (also due to the tumor localization, far from eloquent structures). Regular audiological, endocrinological, and ophthalmological evaluations were always found to be normal. Neurological physical examination never revealed alterations. Currently, the girl is attending her second year of secondary school with good academic performance. Even from the neuropsychological evaluations, no problems were ever highlighted.

## 4. Conclusions

RGNT is a rare, low grade, central nervous system tumor. The mainstay of treatment is surgery and usually no adjuvant therapy is necessary. Given the rarity and the few described cases of recurrences, there is no clear evidence or indication for its management. We described a pediatric case of a patient who experienced two local recurrences, the first retreated with surgery and the second with proton beam therapy, achieving complete remission, and good clinical conditions highlighted at the four-year follow-up. The physical properties of protons (useful in reduction of radiation field and exit dose, compared with typical photon treatment) are appropriated for localized low-grade gliomas and should be considered as adjuvant therapy in recurrences.

## Figures and Tables

**Figure 1 diagnostics-11-00903-f001:**
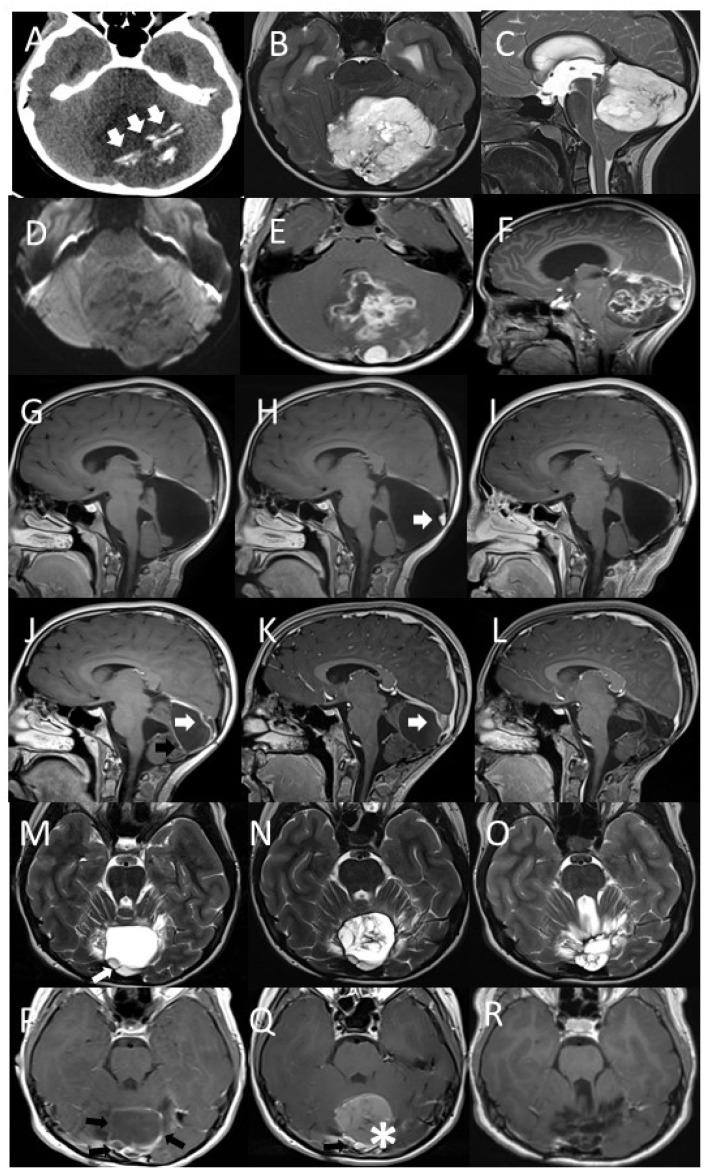
Axial CT image (**A**), axial (**B**), and sagittal (**C**) T2w MRI images show a well-demarcated cerebellar mass, compressing fourth ventricle, with intratumoral mineralization (arrows), elevated diffusion coefficient (**C**) indicating low cellularity. Axial (**D**) and sagittal (**E**) Gd T1w images show inhomogeneous contrast enhancement. Sagittal Gd T1w images show complete tumor excision (**G**), parietal nodule of the surgical cavity with contrast-enhancement in relation to disease recurrence (**H**, arrow), completely removed at second surgery (**I**). Sagittal Gd T1w images show new relapse of pathology (**J**) with marginal contrast-enhancement of the surgical cavity (black arrow) and peripheral nodular aspects (white arrow); the latter slightly increased at early post-proton therapy control (**K**, white arrow). The last follow-up shows complete disappearance of these pathological findings (**L**). Axial T2w (**M**,**N**,**O**) and GdT1w (**P**,**Q**,**F**) images show new relapse of pathology with peripheral nodular aspects (**M**, white arrow) and marginal contrast-enhancement of the surgical cavity (**P**, black arrows). Early post-proton therapy follow-up shows slight increased contrast-enhancement (**Q**, Black arrow) along with hemorrhagic distension of the surgical cavity (**Q**, star). The last follow-up shows complete disappearance of these pathological findings and tissue distortion with foci of malacia (**R**).

**Figure 2 diagnostics-11-00903-f002:**
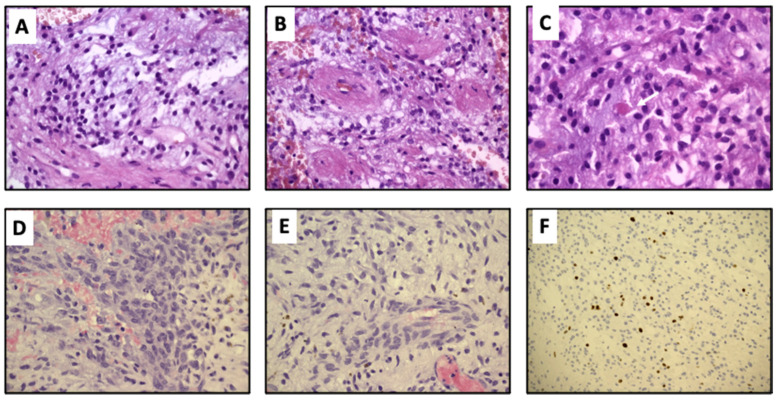
Histology. (**A**) Primary tumor: neurocytic component consisting of uniform medium-sized cells surrounded by a delicate neuropilic stroma (H&E 40×). (**B**) Primary tumor: perivascular distribution around hyaline small vessels (H&E 40×). (**C**) Primary tumor: glial component with pilocytic astrocytoma features; scattered eosinophilic granular bodies were present (arrow) (H&E 63×). (**D**) Recurrence: higher cellularity of the recurrence that showed mild/moderate cytological atypia (H&E 40×). (**E**) Recurrence: glial component was predominant; papillary structures were not observed (H&E 40×). (**F**) Proliferation index was around 4–5% (IHC for Ki67 20×). Primary tumor showed very low proliferation (data not shown).

**Figure 3 diagnostics-11-00903-f003:**
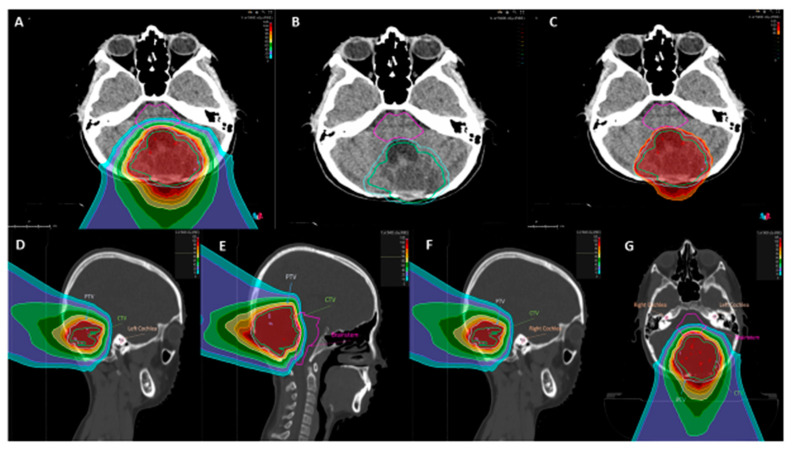
PBT: (**A**,**G**) axial dose distributions of the 3-fields proton plan. (**B**) The clinical target volume (CTV, in light green), the planning target volume (PTV, in light blue) and the brainstem (in purple) contoured are highlighted. (**C**) Reports a focus on the high dose region (>95% of the prescription dose), showing a high dose conformity to the target and homogeneity. Dose brain steam: 52.6 Gray (RBE) left and right cochlea: no dose; 0 Gray (RBE). (**D**–**F**): sagittal dose distribution of the 3-fields proton plan.

## Data Availability

Not applicable.

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
