# Peer review of "Rosette-Forming Glioneuronal Tumor of the Fourth Ventricle: A Case of Relapse Treated with Proton Beam Therapy"

_diagnostics, 2021, doi:10.3390/diagnostics11050903_

Round 1
Reviewer 1 Report
The authors point out the need of treatment in addition to surgery in case of recurrence of this type of tumors (RGNTs) and propose the use of proton beam therapy as an alternative to the typical radiation therapy, which would avoid or reduce its side effects. I think this case report is relevant and important for the audience to read and to take into consideration in the future. I would accept the manuscript after minor revision, since I missed a short description of the proton beam therapy in the introduction, including a short mention to the advantages of proton beam therapy against typical photon treatment, although they are well described in the discussion.
Author Response
PBT offers dosimetric advantages over photon radiation therapy due to steep dose fall off at depth. Decreased integral dose to normal brain and superiority in meeting surrounding tissue constraints has established the role of PBT in management of various central nervous system (CNS) tumors. Studies have demonstrated the dosimetric advantages of PBT, allowing for reduction in radiation-induced toxicities such as neurocognitive decline and secondary malignancy
Reviewer 2 Report
This is a case report of a child with Rossette forming glioneuronal tumor which recurred following gross total resection and was treated with proton beam therapy with good results. There is a review of the literature of the entity, the current treatment and prognosis.
The main weakness of the paper is that there is no molecular workup. This tumor was more aggressive than expected - was it different in some ways from the conventional, previously described Rosette forming glioneuronal tumors? It would have been a much more interesting paper if molecular and methylation data would have been provided as well. As it is - it is really a review article with little novel information.
The pathology description needs a little modification - the opening is that the tumor is biphasic but the description is limited to the neuronal component only. There also should be some English editing - the term "poorly cellulated" should be replaced with "hypocellular" and "elements" should be replaced with "cells".
Overall, a nice paper, review of a rare entity. Not much novel information and sadly no molecular data.
Author Response
The main weakness of the paper is that there is no molecular workup
- We thank the reviewer very much for this right observation; actually we are working on the molecular part and on the methylation profile to better characterize the tumor. In the time required, for this case report, we are unable to define the molecular part. The pathology description needs a little modification - the opening is that the tumor is biphasic but the description is limited to the neuronal component only.
- The glial cells consisted of spindle- or stellate-shaped astrocytic cells with elongated to oval nuclei forming a compact fibrillar meshwork with occasional Rosenthal fibers and oligodendroglial-like cells with round nuclei and clearly staining cytoplasm were present
Reviewer 3 Report
In this manuscript, Cacchione et al report a case of a 9-year-old girl successfully treated for a second post-operative rosette-forming glioneuronal tumor (RGNT) recurrence with proton beam therapy (PBT).
This clinical case is well presented with a clear iconography including successive MRI, and histological images and PBT dosimetric images.
However, images of the second recurrence could be more detailed with axialGd MRI images in figure 1.
Description of dose distributions needs to add sagittal reconstructed CT images with isodoses in figure 3, and doses delivered to cochleas and brainstem.
The first part of discussion is a short review of our current knowledge on RGNT in terms of clinical presentation, radiological aspects, pathological characteristics, and molecular genetic features. It is not clear concerning BRAF : « no BRAF alteration, fusion or mutation have been demonstrated » in lines 147,148 , and, a little later in the text : « KIAA1549/BRAF gene fusion » in line 152.
The second part highlights that, apart from surgery which remains the mainstay of treatment, the indications of chemotherapy and radiotherapy as essentially adjunctive treatments are not clearly validated. In addition, advantages of PBT are recalled in paediatric radiotherapy.
In last part, authors highlight the persisting complete remission at 3 years following PBT without neurlogical side-effects. However, it should be noted that , in this case, excellent CNS tolerance is due essentially to CTV location in posterior fossa, away from critical structures.
Finally, given the rarity and the few described cases of RGNT recurrences, there is no validated strategy and, in this context, this report suggests that PBT could be a safe and efficient treatment .
In conclusion, this report is of interest in that it proposes a non surgical and potentially efficient treatment for RGNT recurrence .
Author Response
- We added axial Gd MRI images in figure 1.
- We added sagittal reconstructed CT images with isodoses in figure 3, and doses delivered to cochleas and brainstem. Dose Brainsteam: 52,6 Gray (RBE) Left and Right Cochlea : no dose ; 0 Gray ( RBE)
- In the discussion we only have mentionated the possibility to find KIAA1549/BRAF fusion in the patogenesis of this tumor
- We inserted in the discussion that the absence of neurological deficits is due to the tumor localization, far from eloquent stuctures